# Investigating the Gut Microbiota Composition of Individuals with Attention-Deficit/Hyperactivity Disorder and Association with Symptoms

**DOI:** 10.3390/microorganisms8030406

**Published:** 2020-03-13

**Authors:** Joanna Szopinska-Tokov, Sarita Dam, Jilly Naaijen, Prokopis Konstanti, Nanda Rommelse, Clara Belzer, Jan Buitelaar, Barbara Franke, Esther Aarts, Alejandro Arias Vasquez

**Affiliations:** 1Department of Psychiatry, Radboudumc, Donders Institute for Brain, Cognition and Behaviour, 6525 GA Nijmegen, The Netherlands; Joanna.Szopinska-Tokov@radboudumc.nl (J.S.-T.); Nanda.Lambregts-Rommelse@radboudumc.nl (N.R.); Barbara.Franke@radboudumc.nl (B.F.); 2Department of Cognitive Neuroscience, Radboudumc, Donders Institute for Brain, Cognition and Behaviour, 6525 EN Nijmegen, The Netherlands; Sarita.Dam@radboudumc.nl (S.D.); jilly.naaijen@donders.ru.nl (J.N.); Jan.Buitelaar@radboudumc.nl (J.B.); 3Laboratory of Microbiology, Wageningen University, 6708 WE Wageningen, The Netherlands; prokopis.konstanti@wur.nl (P.K.); clara.belzer@wur.nl (C.B.); 4Karakter Child and Adolescent Psychiatry University Center, 6525 GC Nijmegen, The Netherlands; 5Department of Human Genetics, Radboudumc, Donders Institute for Brain, Cognition and Behaviour, 6525 GA Nijmegen, The Netherlands; 6Centre for Cognitive Neuroimaging, Donders Institute for Brain, Cognition and Behaviour, Radboud University, 6525 EN Nijmegen, The Netherlands; esther.aarts@donders.ru.nl

**Keywords:** gut microbiota, ADHD, 16S rRNA gene, Inattention

## Abstract

Attention-deficit/hyperactivity disorder (ADHD) is a common neurodevelopmental disorder. Given the growing evidence of gut microbiota being involved in psychiatric (including neurodevelopmental) disorders, we aimed to identify differences in gut microbiota composition between participants with ADHD and controls and to investigate the role of the microbiota in inattention and hyperactivity/impulsivity. Fecal samples were collected from 107 participants (N_ADHD_ = 42; N_controls_ = 50; N_subthreholdADHD_ = 15; range age: 13–29 years). The relative quantification of bacterial taxa was done using 16S ribosomal RNA gene amplicon sequencing. Beta-diversity revealed significant differences in bacterial composition between participants with ADHD and healthy controls, which was also significant for inattention, but showing a trend in case of hyperactivity/impulsivity only. Ten genera showed nominal differences (*p* < 0.05) between both groups, of which seven genera were tested for their association with ADHD symptom scores (adjusting for age, sex, body mass index, time delay between feces collection and symptoms assessment, medication use, and family relatedness). Our results show that variation of a genus from the *Ruminococcaceae* family (*Ruminococcaceae_UCG_004*) is associated (after multiple testing correction) with inattention symptoms and support the potential role of gut microbiota in ADHD pathophysiology.

## 1. Introduction

Attention-deficit/hyperactivity disorder (ADHD) is a common, highly heritable [1], heterogeneous neurodevelopmental disorder with around 5% prevalence in children and 2.5% prevalence in adults worldwide [2]. The disorder is characterized by age-inappropriate levels of inattention and/or hyperactivity and impulsivity. ADHD has a significant social impact on patients’ lives, causing disruption at school [3], work [4], and in personal relationships [5]. Typically, ADHD has its onset in childhood and can persist into adulthood. ADHD is considered a multifactorial disorder with multiple (common and rare) genetic variants, in combination with the environment, explaining its etiology and phenotypic variation [6].

Treatment of ADHD usually involves prescription of either stimulant or non-stimulant medication that target specific systems related to dopamine, noradrenaline, and/or serotonin neurotransmission [7]. These pharmacological interventions are highly effective in controlling ADHD symptoms and have an approximate response rate of 70% [8]. However, medication treatment of ADHD is limited by low adherence, concerns about side effects, and absence of evidence for long-term efficacy [9,10]. Approximately 30% of individuals with ADHD do not respond to medication or are unable to tolerate the adverse effects [7]. For (some of) those patients, non-pharmacological treatments are a suitable alternative [11].

One of the non-pharmacological interventions recently suggested to influence ADHD symptom severity is diet (e.g., elimination diet) [12,13]. Diet can exert its effects on ADHD through the gut-brain axis. The gut-brain axis is a continuous and bidirectional communication system between the enteric and central nervous systems including the cognitive and emotional centers of the brain [14]. The gut-brain axis has been suggested to modulate the risk for several psychiatric illnesses, including ADHD [15,16], and can influence cognitive processes, mood, and brain performance [17].

A key player in the gut-brain axis is the complex ecosystem of commensal bacteria living in our gut, the microbiota [18]. DNA sequencing makes it possible to investigate the microbial composition and its potential role(s) in the risk and pathophysiology of ADHD [19]. For example, in our previously published study, we observed significantly enhanced predicted microbial biosynthesis of phenylalanine (a dopamine precursor related to an increase in the genus *Bifidobacterium*) associated with a neural hallmark of ADHD, for instance, decreased functional responses of the ventral striatum during reward anticipation [19].

In the present study, we increased the sample size of Aarts et al. (16) (samples overlap ~40%) in order to investigate (i) if there are differences in gut microbiota composition between participants with ADHD and age matched controls, and if so, (ii) whether these bacteria are associated with the severity of ADHD symptoms.

## 2. Material and Methods

### 2.1. Study Participants

To evaluate gut microbiota composition in participants with ADHD and healthy controls, we collected fecal samples from 107 Dutch (Caucasian) individuals enrolled in the follow-up of the NeuroIMAGE study [20]. Three groups were included: participants with ADHD (*N* = 42), subthreshold ADHD (*N* = 15; participants who did not reach the criteria for being considered as ADHD but scored too high to be considered healthy control; since we divided the analysis between case/control and continuous analysis, the subthreshold ADHD group was excluded from case–control comparisons), and healthy controls (*N* = 50). The cohort included sibling pairs, which was taken into account in the analysis. A semi-structured diagnostic interview of DSM-IV criteria was conducted with both the participant and his/her parents using the Kiddie-Schedule for Affective Disorders and Schizophrenia (K-SADS) according to DSM-IV criteria. Clinical diagnosis was confirmed using a diagnostic algorithm which combined the diagnostic interview (K-SADS) with the Conners rating scales [20].

Continuous measures of inattention severity (IA) and hyperactivity/impulsivity severity (HI) were derived from the Conners Adult ADHD Rating Scales (CAARS; ≥ 16 years) and Conners Teacher Rating Scale (CTRS; < 16 years). For participants using medication, ratings were based on each participant’s functioning off medication. Detailed recruitment and diagnostic information can be found in the NeuroIMAGE design article [20].

Additionally, the following information was obtained from medical records: age, sex, Body Mass Index (BMI), time delay between feces collection and symptom assessment (differences in days, further on called “diff_days”), and use of ADHD-related medication. Five samples with missing BMI values were excluded from the regression analysis. For an overview of the participant characteristics, see Table 1. Information regarding the use of ADHD medication was provided via self-report (≥16 years) or parental report (<16 years) on the day of measurement. Two controls were removed because they indicated the use of ADHD medication.

Ethical approval for the study was obtained from the local research ethics committees (Commissie Mensgebonden Onderzoek (CMO) Regio Arnhem-Nijmegen) on 9 April 2013 and can be found under registration number 2012/542 and ethics protocol number NL nr.: 41950.091.12. A written informed consent was obtained from all participants and/or their parents prior to the sample and data collections.

### 2.2. Microbiota Methods and Measures

#### 2.2.1. Sample Collection, Preparation and Sequencing

The human fecal samples were collected at home by the participants and stored at 4 °C. Within 24 hours after collection, the samples were transported to the laboratory, aliquoted into 1.5 ml Eppendorf tubes, and stored at -80 °C. The bacterial DNA was extracted using a repeated bead-beating step and the Maxwell® 16 Instrument (Promega, Leiden, The Netherlands), as described previously [21]. DNA purification was performed with a customized kit (AS1220; Promega, Leiden, The Netherlands). The purified bacterial DNA was measured with a NanoDrop ND-2000 spectrophotometer (Thermo Fisher Scientific, Wilmington, DE, USA), and aliquots of 20 ng/µL were prepared for the 2-step Polymerase Chain Reaction (PCR) reactions (including negative controls). In the first PCR, amplification of the V1-V2 region of the 16S rRNA gene was performed using previously reported primers for this region: 27F-DegS (5’GTTYGATYMTGGCTCAG) – 338RI-II (5’ GCWGCC[T/A]CCCGTAGG[A/T]GT) [22]. In the second PCR, unique barcoded primers were added to each sample to allow for parallel sequencing of many different samples. The PCR product was checked using electrophoresis and purified using the CleanPCR kit (CleanNA, Alphen aan den Rijn, The Netherlands). The DNA concentration was measured using Qubit® 2.0 fluorometer. The purified samples were used to prepare libraries for the Illumina HiSeq PE300 sequencing platform (GATC Biotech AG, Konstanz, Germany), with final loading concentrations of 200 ng/µL.

#### 2.2.2. Data Processing

The sequenced data was analyzed through NG-Tax 16S rRNA pipeline at Wageningen University and Research (WUR, Wageningen, The Netherlands) [23]. NG-Tax identified the taxonomy of the samples based on 16S sequences using three core elements: (i) barcode-primer filtering, (ii) operational taxonomic unit (OTU) picking, in which unique sequences with the relative abundance above 0.1% were clustered into OTUs based on a sequence similarity ≥98.5%, and (iii) taxonomic assignment using the SILVA reference database (version 128; [24]).

#### 2.2.3. Filtering Procedure of Taxonomic Data

We performed two filtering steps on the output file (BIOM-file) of NG-Tax in order to remove genera with low prevalence and to reduce the impact of the high number of absent genera (with a value of zero per sample). This step was critical to improve the power to detect the true effects of the microbiota while keeping as much information as possible. The way the genera/OTUs were identified made it impossible to disentangle if the observed values of zero correspond to true zeros (e.g., not present in the sample) or are false zeros (e.g., present but not detected). The two filtering steps were applied as follows: (i) the OTU table was filtered at the genus level, where a genus with non-zero values in less than 10% of the samples was removed, and (ii) the OTU table was filtered at sample level, at which a sample with less than 10% of genera was removed (Appendix A). The results of 16S rRNA analysis after filtering of taxonomic data can be found in the Appendix A.

#### 2.2.4. Sequencing Depth Comparison

The sequencing depth of the microbiota data was compared between all groups (ADHD, controls and subthreshold ADHD) by performing a Kruskal–Wallis H test on the total reads. This was done in order to assess equal distribution of the sequence reads across the groups, which helped to verify the effect of any technical variation between the groups. The results can be found in the Appendix A, which indicated no differences.

#### 2.2.5. Within-Sample Diversity Metrics

Three alpha-diversity metrics were applied on the OTU level: (1) the species richness estimator, counting the observed unique OTUs in each sample [25]; (2) Shannon–Wiener diversity [25] index, which takes into consideration not only the number of observed unique OTUs but also their abundance; and (3) the phylogenetic richness estimator, which estimates microbial diversity across a phylogenetic tree (Faiths’ phylogenetic diversity) [26]. The alpha-diversity metrics were calculated using the ‘alpha_diversity.py’ script in QIIME 1.9.1 [27] and compared between participants with ADHD and controls.

#### 2.2.6. Between-Sample Diversity Metrics

To assesses beta-diversity, we used the weighted UniFrac distance metric, a phylogenetic-based assessment of the difference in overall bacterial community composition at the OTU level [28]. To analyze the beta-diversity, multivariate statistics were conducted using ADONIS and betadisper functions in the R package vegan version 2.5–2 [29,30]. Through ADONIS, we determined if the tested variables (i.e., disease status or symptom counts) influenced beta-diversity [29]. Betadisper measures the variability in OTU composition among groups (here ADHD and controls) [31]. Principal Coordinates Analysis (PCoA) was performed using the R function phyloseq::ordinate to determine and visualize whether there is a clear discrimination of microbial composition between the two groups.

#### 2.2.7. Taxonomic Composition Analysis and Associations with Symptoms

Taxonomic composition of the gut microbiota was investigated at the phylum and genus levels after transforming the sequencing read counts into microbial relative abundance (normalization step). Any unknown taxonomic level (e.g., unknown genus) was assigned to the next highest known taxonomic rank (e.g., family). The composition analysis was calculated using QIIME 1.9.1 with the “summarize_taxa.py” script [27]. Microbiota compositional data are highly skewed given the high number of zeros. We used Linear discriminant analysis Effect Size (LEfSe; https://huttenhower.sph.harvard.edu/galaxy) for statistical analysis and visualization of the results. LEfSe uses non-parametric statistics (which is less sensitive to the extreme values [32]), in our case the Kruskal–Wallis sum-rank test [33], to identify (nominal) statistical differences in the relative abundance of gut microbiota between participants with ADHD and controls. In order to maximize information content and, at the same time, include as many genera as possible, we included only genera with non-zero values in at least 10% of the samples in both groups. To prioritize the selection of candidate taxa without making any claims of association (with ADHD), genera showing nominal statistical differences (*p* < 0.05, uncorrected) were selected for downstream correlation and linear regression analyses.

Linear mixed regression analysis was performed to associate bacterial relative abundance with inattention or hyperactivity/impulsivity score available for all participants (including participants with “subthreshold ADHD”). Models were adjusted for age, sex, BMI, diff_days, and included the family relatedness as random factor; this was done by using the R function lme4::lmer. Given the skewed distribution of the microbial relative abundance prior to the association analyses, we investigated linear regression assumptions and identified and removed extreme and influential samples (outliers). Outliers are known to have a significant effect on the regression model (but not on the non-parametric test [32]). Outliers were defined by Cook’s distance above (4/n) (where n is the number of observations) and Leverage value above (3 × (k+1)/n) (where k is the number of independent variables, in our case k = 5) [34,35]. Cook’s distance identifies influential values, which do not have to be necessarily the extreme ones; these can be identified by Leverage. Therefore, a sample was excluded from the analysis only if it scored above the threshold for both values. Regression analyses were corrected for multiple testing using the false discovery rate (FDR; in total corrected for 14 tests) and indicated as q-values (Q).

#### 2.2.8. Effect of Medication on the Regression Results and on Gut Microbiota Composition

Often ADHD patients are medicated, and some studies show that medication can influence gut microbiota composition [36,37]. Therefore, we explored an effect of ADHD medication on our (regression) results and on gut microbiota composition at the genus level. The regression model could not simply be adjusted for medication due to the large number of non-medicated cases, who do not equate to healthy controls. Thus, the medicated cases (*N* = 19) were removed from the regression model to see how this effects the results.

#### 2.2.9. Correlation Analysis and Multiple Regression with All Selected Genera

The gut microbiota is a highly complex ecosystem of interacting organisms. In order to investigate the (in)dependent effect of the selected genera on symptoms, we investigated their correlation structure and performed multiple regression analysis. The genus-genus correlation was assessed based on Spearman’s rank correlation coefficient. Multiple regression analysis was performed for the same selected genera tested in the univariate models, adjusting for age, sex, BMI, diff_days, and family relatedness as a random factor; the analysis was done without the samples identified as outliers (see above). If not mentioned otherwise, the data were analyzed using IBM SPSS for Windows 25.0 software (IBM Corp., Armonk, NY, USA), and the analysis was preceded by the Shapiro–Wilk’s normality test.

## 3. Results

### 3.1. Subjects Characteristics

The general characteristics of the studied sample are presented in Table 1. Mean age, median BMI, percentage of males and differences in days between fecal collection and ADHD symptom assessment (diff_days) were similar among the two groups. As expected, mean inattention and hyperactivity/impulsivity scores were statistically different between the ADHD and control groups. Out of the 41 participants with ADHD, 19 were using medication for ADHD.

### 3.2. Microbiota Measures

*Within-and between-sample diversity metrics:* None of the three alpha-diversity (within-sample diversity) measures showed significant differences between the ADHD and control groups (Appendix A).

Beta-diversity (between-sample diversity), assessed using betadisper [30], showed that the ADHD group had a smaller variation in the gut microbiota composition, which means a higher taxonomic similarity (within the group) compared to controls (*P* = 0.004; Figure 1 and Appendix A). ADONIS revealed a significant effect of ADHD diagnosis and symptom severity on the variation in the beta-diversity. The variation in beta-diversity was significantly explained by disorder status (*N* = 88; variance explained = 3.2%; *P* = 0.033), and inattention score (IA) (*N* = 102; variance explained = 3.7%; *P* = 0.014), whereas hyperactivity/impulsivity score (HI) was at trend level (*N* = 102; variance explained = 2.4%; (*P* = 0.059). Age, sex, BMI, and medication did not have a significant effect on beta-diversity (Table 2). Additionally, PCoA based on weighted UniFrac distance did not show a clear discrimination of microbial composition between the two groups determined by disorder status (ADHD vs. controls) (Appendix A).

#### 3.2.1. Taxonomic Composition Analysis and Associations with Symptoms

As expected [38], compositional analysis of our samples revealed that *Firmicutes*, *Bacteroidetes*, *Actinobacteria*, *Proteobacteria*, and *Verrucomicrobia*, were the most frequent phyla in our data (Appendix A). There were no significant differences in the relative abundance of any of these phyla between participants with ADHD and controls (Appendix A).

At the genus level, differences in the gut microbiota composition revealed nominal significant case-control differences for ten genera (*p* < 0.05; Figure 2). Of those, nine genera were increased, and one was decreased in participants with ADHD. Based on their prevalence (present in at least 10% of the samples in each group; see methods), seven were selected for downstream association analyses with ADHD symptom scores (inattention and hyperactivity/impulsivity scores) (Appendix A). One genus, *Ruminococcaceae_UCG_004*, was associated (B = 39.291, *P* = 0.002, Q = 0.027; corrected for multiple testing; Table 3) with inattention scores and two other genera showed nominal associations (*p* < 0.05). We did not find any association between tested genera and hyperactivity/impulsivity score (before or after correcting for multiple testing; all *p* > 0.05); therefore, only IA was taken into consideration in further analyses.

#### 3.2.2. Effect of Medication on the Regression Results and on Gut Microbiota Composition

We tested the effect of ADHD medication on the (regression) results by excluding medicated cases (*N* = 19) from the analysis. We found that medication did not influence the association between *Ruminococcaceae_UCG_004* and symptoms of inattention (B = 47.083, *P* = 0.0006 vs results in Table 3). When comparing the medicated (*N* = 19) versus non-medicated (*N* = 22) individuals with ADHD, we found that the genus *Dialister* was increased and that the genus *Phascolarctobacterium* decreased in medicated ADHD (Appendix A). Regarding the *Phascolarctobacterium* results, we had to treat them with caution because of having only three non-zeros values for medicated cases. It is difficult to disentangle whether the observed values of zero correspond to true zeros (e.g., not present in the sample) or of false zeros (e.g., present but not detected).

#### 3.2.3. Correlation Analysis and Multiple Regression with all Selected Genera

Spearman correlation analysis showed that two genera, *Ruminococcaceae_NK4A214_group* and *Ruminococcaceae_UCG_005*, had a strong positive correlation (r > 0.50) with each other. The other genera showed moderate (0.30 > r > 0.50), weak (0.30 > r > 0.10) or no correlation (Appendix A). Due to the variability in the correlation, we carried out a multiple regression analysis including all selected genera in one model (Appendix A) and investigated the unique contribution of the associated genera to inattention symptoms. Regardless of correlation structure, removing one of the strongly correlated taxa (*Ruminococcaceae_NK4A214_group* or *Ruminococcaceae_UCG_005*) did not significantly change the results. After controlling for other bacterial taxa (and age, sex, BMI, diff_days, and a random factor for family relatedness), *Ruminococcaceae_UCG_004* again showed the highest association with inattention score (B = 43.920, *P* = 0.001), followed by *Ruminococcus_2* (B = 1.525, *P* = 0.001). These results suggest an independent effect of these genera on inattention (Appendix A).

## 4. Discussion

In this study, we aimed to determine the differences in gut microbiota composition between individuals with ADHD and controls and the association between the abundance of the selected genera and the severity of ADHD symptoms (inattention and hyperactivity/impulsivity) accounting for the effects of medication.

To our knowledge, our findings constitute the first (proof-of-concept) of its kind showing the association between the microbiome relative abundance and ADHD symptoms in the adolescent/adult group. Our results showed general differences in microbiota composition (beta-diversity) between the groups as well as differences at the genus level, where the relative abundance of the *Ruminococcaceae_UCG_004* genus was associated with ADHD inattention symptoms. Multiple regression analysis suggested that the association between *Ruminococcaceae_UCG_004* and inattention was independent of other (selected) genera. Our results also point towards a potential effect of genera such as *Ruminococcus_2* and *Ruminococcaceae_uncultured* on inattention.

Importantly, our result indicate that ADHD medication did not have an effect on *Ruminococcaceae_UCG_004* association. However, when directly comparing medicated (*n* = 19) and non-medicated individuals (*n* = 22) with ADHD, we found two genera being different: *Dialister* was increased and *Phascolarctobacterium* was decreased in medicated individuals. While this is not the first time that ADHD medication is examined in terms of an effect on gut microbiota [39], it is the first time that gut microbiota composition at the genus level was assessed. However, please note that these (secondary) results should be interpreted with caution given the small sample sizes of the sub-groups.

Our results are in line with the growing evidence that the gut microbiome might be involved in neurodevelopmental disorders. The *Ruminococcaceae* family and the genera belonging to this family have shown altered relative abundance in individuals diagnosed with several psychiatric diseases, i.e., autism, bipolar disorder, anxiety, depression, schizophrenia, as well as ADHD [40,41,42,43]. Even though there is a lack of knowledge about the functional role of *Ruminoccocace_UCG_004* genus and its effect on the host, the *Ruminococcaceae* family is well described. This family is commonly present in the mammalian intestinal tract, displaying the ability to degrade cellulose and hemicellulose from plant material [44]. These compounds are subsequently fermented and converted to short-chain fatty acids (SCFAs), which can be absorbed and used as energy source by the host [45]. SCFAs (acetate, propionate, and butyrate) are produced (among others) by strains of *Ruminococcus* [46], which are known for mucosal colonization [47], and have been shown to play a potential role in autism [48,49] and in ADHD [50]. Furthermore, *Ruminococcus_2*, became significantly associated with symptoms of inattention in the multivariate analysis (Appendix A). This suggests that, even in the absence of significant correlations with other bacteria (Appendix A), there could be a bacterial community interaction pattern that explains more variance than a genus alone on inattention symptoms. Lastly, our lab recently performed a separate study, in which six randomly selected samples from the NeuroIMAGE cohort (same cohort studied here) were used in an animal study of human fecal microbiota transplantation into germ-free wild-type mice [51]. Mice colonized with ADHD gut microbiota had increased anxiety-like behavior and showed significantly altered structural and functional brain characteristics. In this study, *Ruminococcaceae_UCG_004* was positively correlated with a specific anxiety measure in the mice. Taken together, these results suggest that genera belonging to *Ruminococcaceae* family may exert a specific effect on neurobiological processes key to brain development and several psychiatric diseases including ADHD.

As a post-hoc exploratory analysis we blasted the *Ruminococacceae_UCG_004* sequences using NCBI BLAST (https://blast.ncbi.nlm.nih.gov) to query for highly similar sequences. The BLAST search revealed that the sequence of the *Ruminococacceae_UCG_004* genus was included in *Evtepia gabavorous* species (more information, together with the blasted sequences, can be found in Appendix A). *E. gabavorous* seems to be the only isolated human gut bacteria showing ability to consume gamma aminobutyric acid (GABA) [52,53]. This is relevant because GABA is a naturally occurring amino acid that works as a neurotransmitter. GABA is considered an inhibitory neurotransmitter because it blocks, or inhibits, certain brain signals and decreases activity in the central nervous system and plays a role in neuropsychiatric disorders including ADHD [54,55]. Moreover, there are studies pointing out the relevance of studying the gut microbiome in relation to GABA involvement in ADHD etiology [50].

This study should be viewed in the context of several strengths and limitations. Our strengths include the use of a sample with high-quality clinical assessment and of age-matched clinically ascertained controls. The limitations of our study are: i) limited sample size (although it is the largest sample of its kind so far, *N* = 98) and ii) the fact that we were not able to collect information on lifestyle, dietary patterns (including probiotics), or antibiotic use at the time of feces collection. For the former, we applied two QC steps in order to deal with the big number of variables (genera), their expected small effects and big interindividual variation of the gut microbiota. First, we applied an uncorrected non-parametric approach (to identify differences between two groups, reduce number of variables and prioritize the selection of candidate taxa). Second, we applied an outlier detection step, prior to our regression analysis, in order to reduce the chance of false positives/negatives. For the latter, we were only able to collect information on BMI and while we acknowledge that this is not enough to account for the effects of diet and lifestyle, it is encouraging to see that there was no difference between the groups. Moreover, we looked for and removed samples with a very low bacterial diversity (high proportion of zeros) by applying a 10% genus-based frequency cut-off per sample. This can be used as a proxy for those individuals currently using antibiotics which would show a smaller bacterial diversity.

ADHD studies investigating gut microbiota composition show inconsistent results [19,39,43,56,57]. For instance, the study of Aarts et al. (2017) [19] investigated samples that show important overlap (around 40%) with our data set. This paper reported that *Bifidobacterium* genus showed the largest difference (nominally significant) between the ADHD group and controls. We did not replicate this difference in the current study. The reason for the lack of replication between studies could be due to methodological differences. These include DNA extraction [58], 16S rRNA gene region [59], bioinformatic pipeline, data processing and analysis [60], sample size, and study design. Moreover, the age range (children vs. adults) differences between studies can underscore variation in ADHD symptoms, other non-shared environmental influences and gut microbiome composition [61,62]. Follow-up studies (keeping comparable methods and including dietary patterns, comorbid conditions (of ADHD) and bacterial transcriptomics, metabolomics and metagenomics) are needed to replicate the current findings and to understand the complex biological mechanisms underlying our results.

In conclusion, our work showed the differences in microbiota composition between individuals with ADHD and controls, and that variation of a genus from the *Ruminococcaceae* family (*Ruminococcaceae_UCG-004*) is associated with inattention severity. Further studies should validate the present findings and identify potential gut-brain mechanisms via genera belonging to the *Ruminococcaceae* family, such as those related to SCFA production and to alteration of GABA neurotransmitter.

## Figures and Tables

**Figure 1 microorganisms-08-00406-f001:**
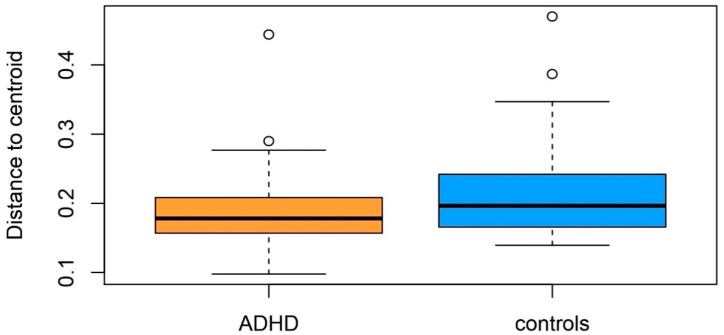
Boxplot of multivariate homogeneity of groups’ dispersions (betadisper) of participants with attention-deficit/hyperactivity disorder (ADHD) and controls. Box plots represent median with whiskers on ± 1.5 IQR. * Pseudo-F = 9.658, *P* = 0.004.

**Figure 2 microorganisms-08-00406-f002:**
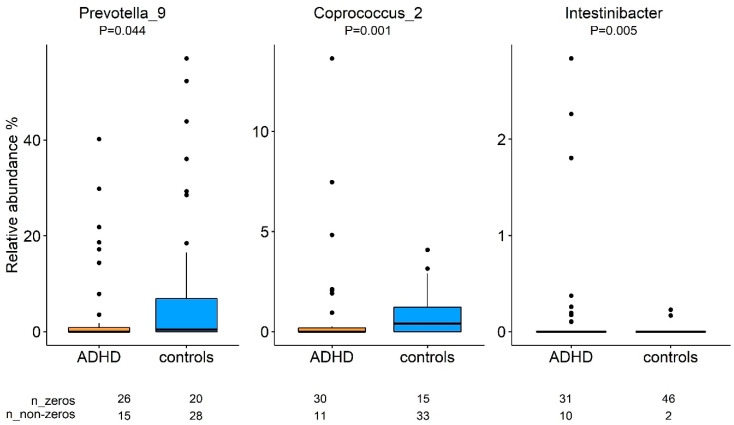
Comparison of bacterial relative abundance between participants with ADHD and controls. Identification of the bacteria differences was done by Kruskal–Wallis test and visualized by LEfSe. Nominal significant threshold: *p* < 0.05.

**Table 1 microorganisms-08-00406-t001:** Characteristics of the sample.

	ADHD	Control	Subthreshold ADHD	*p*-Value ^a^
*N*	41	47	15	-
Age, mean (SD)	20.2 (4.2)	20.5 (3.5)	20.2 (3.3)	NS
Age, range	13–29	13–28	14–26	-
BMI, median (IQR)	23 (20–26)	22 (20–24)	22 (20–25)	NS
BMI, range	16–31	16–31	20–30	-
BMI ≥ 25, %	29	19	20	NS
Male, %	63	49	40	NS
Use of ADHD medication, N	19	0	3	-
Diff_days, median (IQR)	17 (14–34)	30 (12–70)	16 (10–33)	NS
**Conners’**			
Inattention, mean (SD)	66.3 (12.8)	46.8 (12.5)	58.1 (11.5)	<0.001
Hyperactivity/Impulsivity, mean (SD)	59 (12.2)	44.9 (13.0)	59.3 (12.8)	<0.001

^a^ comparison made for ADHD vs. controls; t-test, Mann–Whitney or chi-square test were applied accordingly; one sample had missing value for inattention and hyperactivity/impulsivity scores; four samples had missing value for BMI; NS = not significant; SD = standard deviation; IQR = interquartile rang; diff_days = represents differences in days between fecal collection and Conner’s assessment; - = no comparison was performed.

**Table 2 microorganisms-08-00406-t002:** Beta-diversity analysis.

Variable	*N*	R^2^	Pseudo-F	*p*-Value
Disorder status	88	0.032	2.85	0.033
Age	103	0.004	0.41	0.853
Sex	103	0.011	1.10	0.297
BMI	98	0.005	0.44	0.848
IA	102	0.037	3.87	0.014
HI	102	0.024	2.45	0.059
medication	41	0.022	0.86	0.483

Results of ADONIS on weighted UniFrac dissimilarity matrix including six tests for: disorder status, age, sex, BMI, Inattention (IA), and Hyperactivity/Impulsivity (HI) variables; R^2^ = variance explained, a measure of effect size; Pseudo-F = indicator of the number of clusters, the larger pseudo-F value the greater between-group variation than the within-group variation.

**Table 3 microorganisms-08-00406-t003:** Association of the selected ^a^ genera with ADHD symptoms scores.

	Inattention	Hyperactivity/Impulsivity
	*N*	B (S.E.) ^b^	95% CI	*p*-Value	*N*	B (S.E.) ^b^	95% CI	*p*-Value
*Clostridiales_g__*	97	−1.467 (3.077)	−7.787–5.874	0.634	98	−3.125 (2.439)	−7.922–2.234	0.204
*Family_XIII_AD3011_group*	98	5.323 (2.779)	−0.145–11.483	0.059	98	0.316 (2.709)	−4.879–6.004	0.907
*Ruminococcaceae_UCG_005*	97	1.495 (1.647)	−1.759–4.986	0.367	97	0.175 (1.610)	−2.904–3.384	0.914
*Ruminococcus_2*	98	1.098 (0.445)	0.246–1.959	0.016	98	0.572 (0.440)	−0.268–1.429	0.197
*Ruminococcaceae_uncultured*	96	12.241 (5.011)	2.619–22.264	0.017	93	9.996 (9.191)	−7.132–28.016	0.279
*Ruminococcaceae_NK4A214_group*	97	3.392 (1.860)	−0.230–7.339	0.071	98	2.428 (1.522)	−0.513–5.512	0.114
*Ruminococcaceae_UCG_004*	93	39.291 (12.296)	15.329–64.513	0.002 *	93	12.324(12.147)	−10.849–36.385	0.313

Linear mixed regression models for the relative abundance of the selected genera from the LEfSe pipeline with the ADHD symptoms scores (inattention & hyperactivity/impulsivity) measured from participants with ADHD and controls and subthreshold ADHD; ^a^ The selection of the genera was done prior to regression analysis and it was done based on their prevalence (see the method section); ^b^ Linear mixed regression model without samples removed based on Cook’s distance and Leverage threshold; models adjusted for age, sex, BMI, diff_days, and a random factor for family relatedness; * Significant associations after multiple testing correction (FDR); *N* = number of samples after the removal of outliers (*N* = 98 means no outliers were removed); B = coefficient; S.E. = standard error; CI = Confidence Interval.

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
