# Peer review of "Investigating the Gut Microbiota Composition of Individuals with Attention-Deficit/Hyperactivity Disorder and Association with Symptoms"

_microorganisms, 2020, doi:10.3390/microorganisms8030406_

Round 1
Reviewer 1 Report
Szopinska-Tokov et al characterized gut microbiota composition in individuals with ADHD. The authors identified ten genera whose abundance was significantly different between patients and healthy controls. There is already some literature (not a lot) on the gut microbiome in patients with ADHD and unfortunately I do not think the current work has significantly advanced the knowledge in this field. While the authors claimed that this was the “first” showing an association between microbiome abundance and ADHD symptoms (page 9 lines 14-15), merely comparing bacterial abundance offered very limited insights into the relationship between gut microbiota and the observed phenotypes. As the authors pointed out, more data are needed to support and help understand the microbiome data (page 10 lines 28-29). To me, these are not “follow-up studies” but data that are needed in the current manuscript to warrant a good publication.
Specific comments:
- I do not think the data “suggest a role of gut microbiota in ADHD pathophysiology” as concluded in the abstract. This work only reported correlation data.
- The microbiota composition analysis primarily focused on genera. Given that only the V1-V2 region was sequenced, how many (or percentage) of all the bacteria could be classified at the genus level?
- How were the data from the “subthreshold ADHD” group used? For example in Figures 1 and 2 there were only the patient and control groups.
- The authors may wish to consider using a PCoA plot rather than a boxplot to present the data in Figure 1 to show how the data points were clustered (or otherwise).
Author Response
Response to Reviewer 1 Comments
Point 1: Szopinska-Tokov et al characterized gut microbiota composition in individuals with ADHD. The authors identified ten genera whose abundance was significantly different between patients and healthy controls. There is already some literature (not a lot) on the gut microbiome in patients with ADHD and unfortunately I do not think the current work has significantly advanced the knowledge in this field. While the authors claimed that this was the “first” showing an association between microbiome abundance and ADHD symptoms (page 9 lines 14-15), merely comparing bacterial abundance offered very limited insights into the relationship between gut microbiota and the observed phenotypes. As the authors pointed out, more data are needed to support and help understand the microbiome data (page 10 lines 28-29). To me, these are not “follow-up studies” but data that are needed in the current manuscript to warrant a good publication.
Response 1: We would first like to thank the reviewer for the time spent of reviewing our manuscript and comments helping us improving the article. Our results are indeed based on association analysis and, therefore, the interpretations over the mechanistic insights of our findings are limited. It is important to point out that the aim of our study is to identify differences in the microbial composition between individuals with ADHD and controls. This approach has the potential to provide novel association signals that can be later replicated in bigger samples, which will warrant deep mechanistically research for these insights.
We wanted to follow the reviewer’s advice and in order to elaborate on the potential functionality of our results. We examined the potential functional role of the Ruminociccaceae_UCG_004 taxa which showed an association with inattention score, we added the following paragraph into the discussion and supplementary results:
The fragment added into the discussion in the main text (L50-09, page 9-10):
As a post-hoc exploratory analysis we blasted the Ruminococacceae_UCG_004 sequences using NCBI BLAST (https://blast.ncbi.nlm.nih.gov) to query for highly similar sequences. The BLAST search revealed that the sequence of the Ruminococacceae_UCG_004 genus was included in Evtepia gabavorous species (more information, together with the blasted sequences, can be found in supplementary results in supplementary materials). E. gabavorous seems to be the only isolated human gut bacteria showing ability to consume gamma aminobutyric acid (GABA) [1, 2]. This is relevant because GABA is a naturally occurring amino acid that works as a neurotransmitter. GABA is considered an inhibitory neurotransmitter because it blocks, or inhibits, certain brain signals and decreases activity in the CNS and plays a role in neuropsychiatric disorders including ADHD [3, 4]. Moreover, there are studies pointing out the relevance of studying the gut microbiome in relation to GABA involvement in ADHD etiology [5].
The fragment added into the supplementary results in the supplementary materials file:
The Ruminococacceae_UCG_004 genus was represented by two OTUs/sequences: OTU_65143327 and OTU_65143327 with the total reads of 1555 and 53 respectively. Since the Ruminococacceae_UCG_004 genus was mainly represented by OTU_65143327, the sequences of this OTU were subjected to NCBI BLAST (https://blast.ncbi.nlm.nih.gov) to query for highly similar sequences (the blasted sequences can be found below this paragraph). BLAST indicated that Ruminococacceae_UCG_004 could be Evtepia gabavorous (Accession: MH636586.1), which we confirmed by re-running the NG-Tax pipeline increasing the read length to 100bp (of each forward and reverse), and we revealed that our targeted sequence is in 100% similar to N=11 uncultured bacteria and one cultured bacteria - Evtepia gabavorous (Query Cover: 100%; E-value: 2e-43; Percent Identity: 100%). Moreover, all the uncultured bacteria align with the sequence of E. gabavorous (N=10/11 have Percent Identity of 99-100% and N=1/11 has Percent Identity of 94%). This is probably due to the fact that these sequences were deposited in the NCBI before E. gabavorous was isolated hence named uncultured.
The blasted sequences are:
>OTU 65143327; g__Ruminococcaceae_UCG-004 ; Forward sequence
GATGAACGCTGGCGGCGTGCTTAACACATGCAAGTCGAACGGAGGACCCTTGACGGAGTTTTCGGACAAC
GGATAGGAATCCTTAGTGGCGGACGGGTGA
>OTU 65143327; g__Ruminococcaceae_UCG-004 ; Reverse sequence
CTGGGCCGTGTCTCAGTCCCAATGTGGCCGGTCAACCTCTCAGTCCGGCTACTGATCGTCGCCTTGGTAG
Point 2: I do not think the data “suggest a role of gut microbiota in ADHD pathophysiology” as concluded in the abstract. This work only reported correlation data.
Response 2: We agree with the reviewer that the conclusion can be improved; therefore, we adjusted the conclusion according to the reviewer suggestion (L32 page 1; L26 & L47 page 9; L39-44 page 10). We emphasized that the results are based on association studies, which support a potential role of gut microbiota in ADHD pathophysiology.
Point 3: The microbiota composition analysis primarily focused on genera. Given that only the V1-V2 region was sequenced, how many (or percentage) of all the bacteria could be classified at the genus level?
Response 3: We thank the reviewer for bringing this point. The V1-V2 region has shown reliable results. Chakravorty et al (2007) pointed out that “V2 and V3 were most suitable for distinguishing all bacterial species to the genus level except for closely related Enterobacteriaceae” [6]. Recent paper of Winand et al (2019) showed that a V1-V3 region has very small number of unclassified bacteria [7]. These studies point out the reliability of using V1-V2 region (or a region including V2 fragment) in microbiome studies. However, it is important to notice that a 16S rRNA gene hypervariable regions can be taxa specific [8, 9]. Furthermore, we used up to date bioinformatics pipeline and reference database to keep as much information as possible. Our choice for the V1-V2 region ensured a comprehensive detection and assignment of the present bacterial taxa.
Point 4: How were the data from the “subthreshold ADHD” group used? For example in Figures 1 and 2 there were only the patient and control groups.
Response 4: Indeed, in Figure 1 (page 6) and Figure 2 (page 7) we did not include the subthreshold ADHD group because these figures described the results from the case/control comparisons. As explained in the Material and Methods section (L32-34 page 2), we divided the analysis between case/control and continuous analysis. In our paper, we used the information provided by the individuals from the “subthreshold ADHD” group in the regression analyses between ADHD symptoms and selected bacterial taxa. Please see the Material and Methods section (L32-34 page 2 and L38-39 page 4) and included in Table 3 (page 8).
Points 5: The authors may wish to consider using a PCoA plot rather than a boxplot to present the data in Figure 1 to show how the data points were clustered (or otherwise).
Response 5: As recommended by the reviewer, we used both techniques, boxplot (Figure 1, page 6) as well as PCoA plot (Figure S3 in supplementary materials), for the better representation of the beta-diversity. We selected the boxplot to be in the main text because it shows beta-diversity as a measure of the distance to group centroid allowing to compare the variability of the groups, which is also captured by PCoA. In addition, PCoA was used to see whether there is a clear discrimination of microbial composition between the two groups.
References:
- Strandwitz, P., et al., GABA-modulating bacteria of the human gut microbiota. Nat Microbiol, 2019. 4(3): p. 396-403.
- Strandwitz, P., Neurotransmitter modulation by the gut microbiota. Brain Res, 2018. 1693(Pt B): p. 128-133.
- Novell R, E.-C.S., Rodriguez E., Efficacy and safety of a GABAergic drug (Gamalate B6): effects on behavior and cognition in young adults with borderline-to-mild intellectual developmental disabilities and ADHD. Drugs in Context, 2020. 9: 212601.
- Schur, R.R., et al., Brain GABA levels across psychiatric disorders: A systematic literature review and meta-analysis of (1) H-MRS studies. Hum Brain Mapp, 2016. 37(9): p. 3337-52.
- Dam, S.A., et al., The Role of the Gut-Brain Axis in Attention-Deficit/Hyperactivity Disorder. Gastroenterology Clinics, 2019.
- Chakravorty, S., et al., A detailed analysis of 16S ribosomal RNA gene segments for the diagnosis of pathogenic bacteria. Journal of microbiological methods, 2007. 69(2): p. 330-339.
- Winand, R., et al., Targeting the 16S rRNA Gene for Bacterial Identification in Complex Mixed Samples: Comparative Evaluation of Second (Illumina) and Third (Oxford Nanopore Technologies) Generation Sequencing Technologies. International Journal of Molecular Sciences, 2019. 21(1): p. 298.
- Jumpstart Consortium Human Microbiome Project Data Generation Working, G., Evaluation of 16S rDNA-based community profiling for human microbiome research. PloS one, 2012. 7(6): p. e39315-e39315.
- Chen, Z., et al., Impact of Preservation Method and 16S rRNA Hypervariable Region on Gut Microbiota Profiling. mSystems, 2019. 4(1): p. e00271-18.
Reviewer 2 Report
The authors of the present study investigated the alteration of gut microbiota composition in individuals with attention-deficit/hyperactivity disorder (ADHD). The RNA sequence data of bacterial 16S ribosomal RNA revealed that significant differences in bacterial composition between ADHD and control participants. The authors concluded that variation of a genus from the Ruminococcaceae family is associated with AHDH symptoms.
The study was very well designed, executed and results were interpreted accordingly. I have only minor suggestion. 1) This is an association study where authors studied alteration of gut-microbiota composition and ADHD symptoms. However, in different parts of the manuscript, the authors mentioned that potential role of gut-microbiota in ADHD. I would suggest revise the abstract and discussion section accordingly
Author Response
Response to Reviewer 2 Comments
The authors of the present study investigated the alteration of gut microbiota composition in individuals with attention-deficit/hyperactivity disorder (ADHD). The RNA sequence data of bacterial 16S ribosomal RNA revealed that significant differences in bacterial composition between ADHD and control participants. The authors concluded that variation of a genus from the Ruminococcaceae family is associated with AHDH symptoms.
Point 1: The study was very well designed, executed and results were interpreted accordingly. I have only minor suggestion. 1) This is an association study where authors studied alteration of gut-microbiota composition and ADHD symptoms. However, in different parts of the manuscript, the authors mentioned that potential role of gut-microbiota in ADHD. I would suggest revise the abstract and discussion section accordingly.
Response 1: We would first like to thank the reviewer for the time spent of reviewing our manuscript and the kind words acknowledging the quality of our work. We agree with the reviewer that our work is an association study and that only limited interpretations can be drawn in terms of mechanistic insights. We adjusted it according to the reviewer suggestions and we updated the manuscript to reflect this change (L32 page 1; L26 & L47 page 9; L38-43 page 10). We emphasized that the results are based on association studies, which support a potential role of gut microbiota in ADHD pathophysiology. Additionally, we examined the potential functional role of the Ruminociccaceae_UCG_004 taxa which showed an association with inattention score, and we added a paragraph into the discussion (L50-09, page 9-10) and supplementary results.